# Synephrine and Its Derivative Compound A: Common and Specific Biological Effects

**DOI:** 10.3390/ijms242417537

**Published:** 2023-12-15

**Authors:** Svetlana A. Dodonova, Ekaterina M. Zhidkova, Alexey A. Kryukov, Timur T. Valiev, Kirill I. Kirsanov, Evgeny P. Kulikov, Irina V. Budunova, Marianna G. Yakubovskaya, Ekaterina A. Lesovaya

**Affiliations:** 1Research Institute of Experimental Medicine, Department of Pathophysiology, Kursk State Medical University, 305041 Kursk, Russia; dodonovasa@kursksmu.net (S.A.D.); krukovaa@kursksmu.net (A.A.K.); 2Department of Chemical Carcinogenesis, N.N. Blokhin National Medical Research Center of Oncology, 115478 Moscow, Russia; zhidkova_em@mail.ru (E.M.Z.); timurvaliev@mail.ru (T.T.V.); kkirsanov85@yandex.ru (K.I.K.); mgyakubovskaya@mail.ru (M.G.Y.); 3Faculty of Oncology, Ryazan State Medical University Named after Academician I.P. Pavlov, 390026 Ryazan, Russia; 4Laboratory of Single Cell Biology, Russian University of People’s Friendship (RUDN) University, 117198 Moscow, Russia; e.kulikov@rzgmu.ru; 5Department of Dermatology, Northwestern University, Chicago, IL 60611, USA; i-budunova@northwestern.edu

**Keywords:** synephrine, glucocorticoids, selective glucocorticoid receptor activator, inflammation, cancer, CpdA, metabolic disorders, cardiovascular system, beta-adrenergic receptors

## Abstract

This review is focused on synephrine, the principal phytochemical found in bitter orange and other medicinal plants and widely used as a dietary supplement for weight loss/body fat reduction. We examine different aspects of synephrine biology, delving into its established and potential molecular targets, as well as its mechanisms of action. We present an overview of the origin, chemical composition, receptors, and pharmacological properties of synephrine, including its anti-inflammatory and anti-cancer activity in various in vitro and animal models. Additionally, we conduct a comparative analysis of the molecular targets and effects of synephrine with those of its metabolite, selective glucocorticoid receptor agonist (SEGRA) Compound A (CpdA), which shares a similar chemical structure with synephrine. SEGRAs, including CpdA, have been extensively studied as glucocorticoid receptor activators that have a better benefit/risk profile than glucocorticoids due to their reduced adverse effects. We discuss the potential of synephrine usage as a template for the synthesis of new generation of non-steroidal SEGRAs. The review also provides insights into the safe pharmacological profile of synephrine.

## 1. Introduction

This review provides a comparative analysis of two biologically active compounds that have been intensively studied in the past decades both as potential drugs.

Synephrine is found in the fruits of several trees from the Rutaceae family, including bitter orange (*Citrus aurantium*), or Seville orange, sour orange, green orange, immature bitter oranges, also called “Zhi Shi” and “Kijitsu”, as well as from some other citrus species such as varieties of tangerines “Nova” (*Citrus deliciosa* “Nova”) and Marr’s Early sweet orange (*Citrus sinensis*) [1]. Synephrine had been widely used in traditional Chinese medicine as an energy stimulant due to its beneficial effects on the cellular energetic [2]. Subsequently, it was introduced as a biologically active dietary supplement with thermogenic and weight-loss properties [3]. Since ephedrine-containing dietary supplements were banned in April 2004, ephedrine-free dietary supplements including synephrine have become widely available on the market [4]. The impact of synephrine on regulating the metabolic rate can be mainly explained by its molecular effects as a non-specific agonist of β-adrenergic receptors, the well-known regulators of lipolysis [5]. It provided the solid groundwork for using synephrine as a safer analogue of ephedrine, which had been banned in several countries either as a drug or a as biologically active dietary supplement [6]. As a low-molecular-weight compound, synephrine may have multiple targets. Specifically, quantitative proteomics and bioinformatics analyses, as well as in vitro and in vivo experiments, revealed synephrine to regulate several sets of genes involved in tumor development and inflammation [7,8,9,10]. Moreover, the similarity between the chemical structures of synephrine and several biologically active compounds makes it reasonable to explore the synephrine effects and their molecular mechanisms.

The non-steroidal selective glucocorticoid receptor agonist (SEGRA), the molecule of the natural origin, 2-(4-acetoxyphenyl)-2-chloro-N-methyl-ethylammonium chloride, also known as Compound A or CpdA, was first described in the 1990s. CpdA binds the glucocorticoid receptor (GR) with an affinity comparable to that of classic glucocorticoids (GCs) [11]. However, CpdA only partially activates GR, mostly inducing a therapeutically important transrepression (TR) function, a negative protein–protein interaction of GR with a variety of transcription factors (TF) [12,13]. GR binding to TFs such as NF-kB and AP1, among others, suppresses their activity, underlying the anti-inflammatory and anti-cancer effects of GCs and CpdA [14,15,16]. The experimental evidence suggests that GR transactivation (TA) mediated via binding of activated GR to GC-responsive elements (GRE) in the gene promoters results in preferential transcriptional activation of the genes associated with metabolic and atrophic adverse effects of GCs [17,18]. The failure of CpdA to induce full-fledged GR activation has been linked to its non-steroidal structure and consequently its inability to promote the formation of GR-GR homodimers necessary for GR binding to palindromic GREs and the activation of target genes’ transcription [19,20]. Thus, CpdA has been extensively studied as an alternative to GCs for clinical applications. However, CpdA use in clinics is restricted by its chemical instability and decomposition to the intermediate metabolite aziridine, a well-known carcinogen [21,22]. Many attempts were made to find/generate another SEGRA by screening molecular libraries, analysis of the components of anti-inflammatory natural extracts; directional chemical synthesis [23], or by virtual GR docking [24]. Nevertheless, this ~25-year effort by pharmaceutical companies and academia to identify or synthesize a clinically promising alternative has not brought the expected results.

Synephrine is the final product of CpdA metabolism and at the same time it serves as precursor for CpdA synthesis [11,25]. Given the high biological activity of synephrine, the analysis of common and specific targets of CpdA and synephrine becomes particularly relevant. Here, we discuss the biological activity of synephrine, its source, chemical and pharmacological features, target receptors, anti-inflammatory, anti-cancer, and other effects, as well as its possible derivatives as potential clinical benefits in comparison with the effects of GR ligands, selective glucocorticoid receptors agonist CpdA, which are structurally similar to synephrine.

## 2. Source, Chemical and Pharmacological Features of Synephrine

Synephrine is a phenethylamine alkaloid which is 4-(2-aminoethyl)phenol substituted with a hydroxyl group and a methyl group at the amino nitrogen [4]. There are three different positional isomers of synephrine (ortho/-o-, para/-p- and meta/-m-) [5,8]. Their chemical structure is similar to ephedrine, the phenylpropylamine derivative that does not contain a substituted hydroxyl group in the phenyl (benzene) ring (Figure 1) [26].

Importantly, the presence of a hydroxyl group in the synephrine molecule as well as the lack of a methyl group on the side chain modifies its stereochemistry and impairs the interaction with proteins. The latter results in an altered pharmacokinetic profile, allowing for synephrine to permeate the blood–brain barrier. For example, the lipid solubility of p-synephrine is much lower compared to ephedrine, resulting in a decreased transport of p-synephrine to the central nervous system (CNS) [27].

The p-synephrine isomer is widely used as a dietary supplement for weight loss since it promotes fat oxidation [1,6]. The safety of p-synephrine to be used in thermogenic dietary supplements was demonstrated in multiple studies [7,26,28,29].

m-Synephrine, also called phenylephrine, is considered to be the most potent adrenergic agonist of the synephrine at α1-adrenoreceptors (α1-AR) as compared to other isomers [1]. The o-synephrine isomer is not found in dietary supplements, and no pharmacological effect of it on humans has been revealed [30]. Structural and stereochemical differences between p-synephrine and m-synephrine lead to different binding characteristics in relation to AR. For example, weak binding of p-synephrine to AR explains its lower deleterious cardiovascular effects as compared to m-synephrine [31]. In addition, each positional isomer can also be found in two enantiomeric forms varying in pharmacological and physiological activity [1].

Another component of a bitter orange extract with a similar structure is norsynephrine (p-octopamine), an octopamine receptor ligand, a neurotransmitter, and an adrenergic receptor ligand with low affinity [32]. A number of synephrine derivatives (such as isopropylnorsynephrine and methylsynephrine) are thermogenic agents [33,34].

Synephrine is an optically active compound that exists as a mixture of R- and S-isomers. The R-enantiomer is the main form in *Citrus aurantium* (92–96%) [35,36,37,38]. There are several lines of evidence that R-synephrine is a more potent agonist in relation to vascular α-, β1-, and β2-AR [39]. At the same time, S-enantiomer has been suggested to block noradrenaline reuptake [40], which may elevate blood pressure and heart rate. In line with these findings, only S-synephrine has been shown to possess antidepressant-like activity [40]. Of note, CpdA enantiomers slightly differ in their biological effects from a racemic mixture. In our previous studies, we synthesized CpdA enantiomers, and in our in vitro studies S-CpdA was revealed to exert stronger proapoptotic effects on leukemia cells as compared to R-CpdA [41].

## 3. Mechanism of Synephrine Effects

Synephrine is an adrenergic receptor agonist that acts predominantly through the β-adrenergic receptors (β-ARs). β-ARs are a subfamily of G protein-coupled receptors (GPCRs) that are expressed by most cell types in humans [9]. This subfamily consists of three members, β1, β2, and β3-AR, and are the targets of endogenous catecholamines (epinephrine and norepinephrine) [42]. β-Agonists promote receptor-mediated G protein activation of the adenylyl cyclase enzyme and increase cyclic adenosine monophosphate (cAMP) production. Regardless of the subtype differences, all activated β-ARs are typically phosphorylated by regulatory kinases such as GPCR kinases (GRKs), and signaling is then terminated by interaction with β-arrestin. This process is called desensitization [42,43].

p-Synephrine and m-synephrine differ in their biological activity due to certain stereochemical differences as separate isomers bind to receptors with different affinity [31]. Therefore, p-synephrine has a lower ability to stimulate α-1, α-2, β-1, and β2-AR compared to the typical sympathomimetics. Moreover, p-synephrine predominantly binds to α1-AR, with lower affinity to α2-AR and thus a much lower potency relative to β-AR, regardless of the subtype [44]. The in vitro studies described above show that the biological effects of p-synephrine are mediated by various mechanisms different from selective binding to specific ARs and with a limited contribution of α-, β-1, β-2, and β3-AR. Analysis of the chemical structure and pharmacological action of synephrine indicates that its molecular action is similar and involves interference with the 3′,5′-cyclic adenosine monophosphate/protein kinase A (cAMP/PKA) signaling. This pathway is triggered by sympathetic activation to restore cellular homeostasis via stimulating glucose uptake, lipolysis, fatty acid oxidation, mitochondrial biogenesis, and cell proliferation [8,45].

Several studies show that p-synephrine can bind to and activate β3-AR, contributing to thermogenesis, lipolysis, glucose, and cholesterol metabolism, and possibly reducing food intake. β3-AR is a G protein-coupled receptor functionally linked to Gαs and Gβγ subunits. Stimulating β3-AR increases intracellular cAMP levels via adenylyl cyclase activation. Consequently, cAMP triggers two major signaling cascades in adipocytes: protein kinase A (PKA) signaling as well as exchange protein which is directly activated by the cAMP/Ras-related protein (EPAC/RAP) pathway. PKA catalyzes the phosphorylation of hormone-sensitive lipase (HSL) and perilipin, stimulating lipolysis [46,47] and inducing thermogenic gene transcription through activating transcription factor 2/cAMP-responsive element-binding protein (ATF-2/CREB) TA [48,49,50,51]. EPAC functions as guanine nucleotide exchange factor (GEF) for the Ras-like small GTPases, or Ras-proximate proteins, Rap1 and Rap2. EPAC acts synergistically with the cAMP-dependent protein kinase PKA via Rap [52,53].

In invertebrates, p-synephrine binds to octopamine receptor subtypes homologous to vertebrate adrenergic receptors. However, the affinity of p-synephrine, m-synephrine, and norepinephrine binding to octopamine receptors varies considerably and has a non-linear correlation with their AR-binding characteristics. Therefore, p-synephrine affinity to octopamine receptors cannot be extrapolated to human or other vertebrate ARs [54]. There is some evidence that synephrine also has weak affinity to 5-hydroxytryptamine (serotonin) receptors (5-HT2A and 5-HT1D) and may interact with trace amine-associated receptor 1 (TAAR1) [44]. TAAR1 is essential for regulating neurotransmission in dopamine, norepinephrine, and serotonin neurons in the CNS and mediates the mechanisms involved in mood changes, attention, memory, and anxiety disorders. Moreover, TAAR1 has recently been described to mediate or modulate immune dysregulation [55,56]. TAARs represent another mechanism underlying p-synephrine action either as a neurotransmitter precursor or as a neuromodulator [44].

Β-agonists have also been reported to affect GR function. Forskolin, which boosts intracellular cAMP production thus mimicking β2-AR activation, elevates GR expression in rat hepatocellular carcinoma cells and potentiates the production of dexamethasone-induced neurotensin, regulating analgesia, thermoregulation, reward, arousal, blood pressure, and ultimately modulating feeding behaviors and body weight [57]. Forskolin can also antagonize the negative autoregulation of GRs induced by dexamethasone. These effects indirectly suggest that β-receptor agonists may enhance GR function by increasing the intracellular cAMP levels [58].

## 4. Chemical and Pharmacological Features of CpdA and Its Target Receptors

The biological effects of both GCs and the selective glucocorticoid receptor agonist CpdA are mediated by GR, a transcription factor from the superfamily of nuclear hormone receptors [59]. GR regulates protein, lipid, and carbohydrate metabolism and the cell cycle, immune, stress, and inflammatory responses, affecting thousands of genes [60]. The effects of receptor activation upon binding to GC or CpdA are mediated by DNA-independent transrepression (TR), which underlies its therapeutic, anti-inflammatory, and anti-cancer activities. Upon transactivation (TA), GR dimers bind to GC-responsive elements (GRE), followed by a transcriptional activation of the genes, which is associated with an elevated risk of adverse effects [17]. The adverse effects, including diabetes mellitus, peptic ulcer disease, Cushing’s syndrome, osteoporosis, hyperglycemia, skin and muscle atrophy, psychosis, glaucoma, and many other complications, are mainly induced by TA and are partially irreversible [18,61].

CpdA, a well-characterized SEGRA, is a synthetic analog of a natural compound found in *Salsola tuberculatiformis* Botschantzev and exerts GR-dependent anti-inflammatory and anti-cancer effects [16,41,62,63]. CpdA appears to be a promising alternative to GCs with fewer side effects [18,41,51,62,64,65,66].

In water solution CpdA decomposes directly to acetyl synephrine in about 5 days, followed by ester hydrolysis to synephrine after a few weeks or months. In contrast, in buffer solutions, CpdA forms a mixture of optical isomers of aziridine within a few minutes. With a half-life (t_1/2_) of several days, aziridines are subsequently hydrolyzed to acetyl synephrine and then to synephrine [1,22] (Figure 2). Whether synephrine and its derivatives could exhibit the SEGRA activity inducing only GR TR without TA-related adverse effects remains to be elucidated.

It is well known that CpdA acts as a GR ligand. It binds to GR with high affinity, causing moderate GR nuclear translocation, but no GR dimerization is necessary for GR binding to palindromic GRE sequences in GR target gene promoters/enhancers [67]. Furthermore, CpdA does not induce GR phosphorylation at Ser211 and hence does not enhance GR TA activity [68]. In contrast, CpdA and classic GCs have remarkably similar TR profiles, suppressing the activity of many pro-proliferative and anti-apoptotic transcription factors: NF-κB, AP-1, Ets-1, Elk-1, SRF, and NFATc [69]. Apart from its SEGRA function, CpdA acts as an antagonist of the androgen receptor. Although CpdA has lower affinity for the androgen receptor compared to active androgens, CpdA induces considerable androgen receptor nuclear translocation, suppresses the interaction between the NH(2)- and COOH-terminal domains of the androgen receptor that are critical for its function, and inhibits both constitutive and dihydrotestosterone (DHT)-inducible androgen receptor transcription activity [62,70]. CpdA does not significantly affect the activity of other steroid hormone receptors such as mineralocorticoid, progesterone, and estrogen receptors [15].

The beta-adrenergic system plays a crucial role in the body’s response to stress and is involved in various physiological reactions to stress, including increasing cardiovascular output, bronchodilation, and mobilization of energy resources via increased breakdown of glycogen into glucose (glycogenolysis) and via lipolysis breakdown of stored fat into free fatty acids which can be used as an energy source. In turn, GCs that are often referred to as stress hormones also mobilize energy resources and increase glucose blood levels via the induction of protein catabolism and lipolysis needed for gluconeogenesis. This overlap in functions of GCs and beta-adrenergic agonists has inspired efforts to identify the potential cross-talk between downstream signaling mediated by GR and b-AR and even at the level of ligand cross-binding.

A crosstalk between GR and β-AR has been described in the literature based on two mechanisms. GCs regulate β-AR binding to G proteins, followed by adenylyl cyclase activation. There are several phosphorylation pathways, including β-adrenergic kinase (β-ARK), and they represent the first step in the development of drug tolerance. The impact of GCs helps to restore the function of the receptor, returning β-AR to a sensitized state. GCs can also enhance the suppressed β-AR function following chronic agonist exposure. Reduced β-AR function is characterized by the internalization and degradation of β-AR, which requires its rapid resynthesis. Binding of the activated GR to GRE in the promoter region of the β receptor gene increases the gene transcription rate and, hence, the protein level [58]. Synephrine’s effects on GR remain unknown.

On the other hand, it is well documented that in asthma, the clinical efficacy of inhaled GCs is enhanced by *β*_2_-adrenoceptor agonists, and the mechanisms underlying their effects on GR function were extensively studied in bronchial epithelial cells (pHBECs) and airway smooth muscle cells [71,72]. Surprisingly, it was shown that GR-mediated transcription could be significantly increased by *β*-agonists without changes in GR phosphorylation, nuclear translocation, or increased loading on GREs as assessed by ChIP [73]. As many GR-target genes are independently induced by *β*-agonists, gene-specific control by GR- and *β*-agonist-activated transcription factors may explain the increased activation of GR in the presence of GCs and *β*-agonists.

## 5. Overlapping Targets and Activities of Synephrine, SEGRA, and Glucocorticoids in Pathological Conditions

### 5.1. Anti-Inflammatory Effects

The anti-inflammatory effects of synephrine have been studied in various models of inflammation, but a detailed mechanism of action has not been fully established. In lipopolysaccharide (LPS)-stimulated RAW264.7 murine macrophages, p-synephrine inhibited the production of pro-inflammatory cytokines and nitric oxide. This effect was associated with the suppression of p38 mitogen-activated protein kinase (p38 MAPK) and nuclear factor kappa B (NF-κB) signaling pathways and was mediated by β-AR [9]. In the acute lung injury model, the anti-inflammatory effects of p-synephrine were accompanied by a decreased activity of pro-inflammatory tumor necrosis factor-α (TNF-α) and interleukin-6 (IL-6) genes, an increased activity of interleukin-10 (IL-10) and superoxide dismutase (SOD), as well as suppressed reactive oxygen species (ROS) generation, reduced myeloperoxidase activity, and inhibited NF-κB phosphorylation [74,75]. Moreover, synephrine reduced the degradation of the NF-κB inhibitor, a nuclear factor of kappa light polypeptide gene enhancer in the B-cell inhibitor alpha (IκBα) [75]. Under normal physiological conditions, NF-κB is present in the cytoplasm in an inactive form in a complex with IκBα. Intranasal administration of LPS leads to the activation of NF-κB and degradation of IκBα in the lung tissue. However, pre-treatment with p-synephrine prevented these processes in vivo [75,76].

In addition, p-synephrine significantly inhibited the proliferation of neutrophils and macrophages in the bronchoalveolar lavage fluid, leading to an anti-inflammatory effect [75]. In a murine model of systemic inflammatory response, p-synephrine inhibited serum levels of pro-inflammatory cytokines and reduced the inflammation [9]. In studies harnessing NIH/3T3 murine fibroblasts and normal human fibroblasts, p-synephrine dose-dependently inhibited IL-4-induced expression of eotaxin-1 via suppression of a signal transducer and activator of transcription (STAT6). STAT6 is critical for activating cytokine gene expression and cytokine signaling in immune cells and target tissue cells. As eotaxin-1 is a chemoattractant and mediator of eosinophilic inflammation development, p-synephrine also inhibited eosinophil recruitment [26].

Inflammation is strongly associated with oxidative stress. The antioxidant properties of p-synephrine were shown in the murine model of diabetes mellitus developed by alloxan injection. p-Synephrine significantly increased the activity of SOD, catalase (CAT), and glutathione (GSH) and reduced the level of malondialdehyde (MDA) in the blood serum of diabetic mice. In addition, p-synephrine administration inhibited kidney inflammation by downregulating TNF-α, IL-6, and IL-1β gene expression levels. The mechanism of anti-inflammatory effects may include the suppression of NF-κB activation and MAPK phosphorylation [7].

The structural similarity of synephrine and CpdA allows for the proposal of generalized mechanisms of anti-inflammatory effects [6,9]. CpdA exerts its anti-inflammatory function via reducing p65 DNA-binding activity in vivo, as well as via inhibiting NF-κB transactivation potential [14]. Furthermore, CpdA is partially responsible for the GC-dependent inhibition of inflammation by inhibiting the activator protein-1 (AP-1) involved in the transcription of pro-inflammatory cytokine genes [17,48,50,77,78], suppressing MAPK [79,80] and STAT signaling and resulting in a downregulated production of cytokines including IL-1β, IL-2, IL-6, and TNF [58,79], as well as a reduction in the phospholipase A_2_ activity, and a decrease in cyclooxygenase-2 (COX-2) expression [81,82] (Table 1).

Moreover, the regular use of β2-agonists has been suggested to cause adverse effects in asthma control due to the crosstalk between cAMP responsive element binding (CREB) proteins and GRs [83]. The cross-talk between GR and β-AR ligands may rely on the cAMP-dependent activation of the nuclear transcription factor CREB, which further binds to cAMP response elements (CREs) in the promoters of target genes. This involves the competitive binding of GRs and protein cofactors, such as CREB or the related P300, which is required for activating transcription factor response elements. Furthermore, there is evidence for mutual inhibition of nuclear transcription factors by β-AR agonists and GCs due to the interaction between GR and CREB induced by the high concentrations of β2-AR agonists [84] (Table 1).

**Table 1 ijms-24-17537-t001:** Overlapping targets and activities of synephrine and CpdA in pathological conditions.

Possible Molecular Mechanisms	Effects	Model Description	Substance
Anti-inflammatory effects			
In vitro			
Downregulated p38 MAPK and NF-κB signaling pathway; Inhibited expression of pro-inflammatory cytokines: IL-8, IL-6, TNF-α [9]		LPS-stimulated RAW264.7 cells	Synephrine
Inhibited IL-4-induced expression of eotaxin-1 via suppression of STAT6 [26]		NIH/3T3 mouse fibroblasts	
Attenuated expression of TNF-α, iNOS, and IL-1, but increased expression of anti-inflammatory IL-10; Induced macrophage differentiation towards M2 anti-inflammatory phenotype [85]		Immortalized murine macrophage cell line RAW 264.7	CpdA
Inhibited NF-κB activity and IKK phosphorylation; Induced IκB-α accumulation; decreased IL-1β expression [86,87]		Synovial fibroblasts from patients with rheumatoid arthritis	
In vivo			
Reduced TNF-α, IL-6 and increased IL-10 activity; Elevated SOD activity and suppressed ROS generation; Reduced MPO activity; Attenuated histological changes; Inhibited NF-κB phosphorylation and IkB degradation [75]	Inhibited pulmonary edema; Reduced histological changes	LPS-induced ALI, mice	Synephrine
Reduced serum levels of proinflammatory cytokines [9]	Improved survival rate	LPS-induced systemic inflammatory response syndrome, a mouse model	
Increased activity of SOD, CAT, and GSH; Reduced MDA content; Downregulation of TNF-α, IL-6, and IL-1β gene expression levels [7]		The mouse model of diabetes mellitus	
Suppressed NF-κB activity and nuclear translocation; Inhibited STAT6 activity and nuclear translocation; Reduced expression of Th2-cytokines: IL-4, IL-5, and IL-13 [88]	Reduced inflammatory cell infiltration in lungs, cytokine production, mucus, and Ig production; Reduced development of airway hyperresponsiveness	Ovalbumin-induced Th2-driven asthma model	CpdA
Decreased NF-κB activity and downregulation of pro-inflammatory cytokines: IL-8, IL-6, and E-selectin [14]	Decreased swelling	Zymosan-induced inflamed paw	
Inhibited pro-inflammatory cytokines: IL-1β, TNF-α, IL-6; Upregulation of anti-inflammatory cytokines: IL-4 and IL-10 [89]	Protected from the development of diabetes; Modulated peripheral immune response (switching from Th1/Th17 towards anti-inflammatory T-regulatory/Treg response)	Streptozotocin-induced model of type 1 diabetes	
Anti-cancer effects			
In vitro			
Reduced expression of p-AKT, AKT, p-ERK, and ERK [90]	Suppressed proliferation	Esophageal squamous cell carcinoma	Synephrine
Increased ROS formation; Increased activity of the antioxidant molecules glutathione and catalase [8]	Revealed no cytotoxic effect	Human colon adenocarcinoma (Caco-2) cells	
Induced DNA damage and apoptosis; Hyperproduction of intracellular ROS [91,92]		Human hepatocellular carcinoma (HepG2)	
Increased expression of Bax and p53 at the mRNA and protein levels; Suppressed PI3K/AKT/mTOR signaling pathway [10]		Lung cancer cells (H460)	
Inhibited several transcription factors, including nuclear factor kappa B (NF-κB), AP-1, Ets-1, Elk-1; Induced caspase-dependent apoptosis [69]	Decreased growth	Highly malignant androgen-independent DU145 and PC3 cells	CpdA
	Strongly inhibited growth and viability [63]	CEM T-cell acute lymphoblastic leukemia; K562 chronic myeloid leukemia cells	
Inhibited NF-κB signaling [16]		Murine L929sa fibrosarcoma cells	
Increased GR-GR dimerization; Decreased number of MR-GR heterodimers [93]		Rat pheochromocytoma PC12 cells	
In vivo			
Reduced level of glucose metabolism genes, G6Pase, and PEPCK [90]	Reduced glucose production	Human ESCC xenografts in nude mice	Synephrine
Induced apoptosis in cancer cells via the upregulation of pro-apoptotic members of the B-cell lymphoma (Bcl-2) family [93]		P2 rat pups	CpdA
Metabolic and anti-diabetic effects			
In vitro			
Reduced level of glucose metabolism genes, G6Pase, and PEPCK [94]	Reduced glucose production	Rat liver cells (H4IIE)	Synephrine
Acted as partial GR antagonist [95]		Immortalized murine keratinocytes	CpdA
In vivo			
Suppressed gene expression levels of TNF-α, IL-6, IL-1β; Activated enzymes of the antioxidant system; Inhibited oxidative stress via suppressing the NF-κB and MAPK pathways [7]	Prevented alloxan-induced changes in body weight, organ parameters, serum uric acid, and serum creatinine, and improved lipid profile	Alloxan-induced diabetes mellitus in mice	Synephrine
Increased metabolic rate via agonistic activity on β-3 adrenoreceptors; Exhibited hypoglycemic and insulin-stimulating properties; Stimulated translocation of the glucose-4 transporter protein [96]	Decreased blood glucose levels; Increased insulin levels; Decreased insulin resistance	Gliclazide-treated rats and rabbits	
Did not induce gluconeogenesis enzymes in the liver [87]	Did not increase in blood glucose levels; Did not induce hyperinsulinemia	Collagen-induced arthritis in mice	CpdA
Inhibited endogenous GR signaling due to CpdA antagonistic effect on GR activity [95]	Revealed anti-inflammatory and atrophogenic effects	Model of contact dermatitis in mice	

### 5.2. Anti-Cancer Effects

The anti-cancer effects of synephrine are largely unknown. However, several in vitro studies on synephrine have described its anti-cancer activity. Synephrine hydrochloride inhibits the proliferation of esophageal squamous cell carcinoma (ESCC) cells in a dose-dependent manner, significantly suppresses cell migration and invasion, and reduces the growth of human ESCC xenografts in nude mice. In addition, synergistic anti-cancer effects of synephrine with 5-FU, one of the commonly used chemotherapy drugs, are demonstrated in ESCC cells [90].

As described in the pertinent literature, synephrine is involved in the regulation of AKT and extracellular signal-regulated kinase (ERK) signaling. The levels of expression of p-AKT, AKT, p-ERK, and ERK in ESCC cells treated with synephrine are reduced in a dose-dependent manner. Synephrine downregulates galectin-3, an activator of the AKT and ERK signaling pathways which plays a crucial role in cell survival and malignization [97]. Therefore, the studies highlight the therapeutic potential of synephrine as an agent that inactivates AKT and ERK signaling, pathways that have been previously described as targets for anti-cancer therapy [90].

Recent studies in lung cancer cells H460 demonstrate the ability of synephrine to reduce cell viability by increasing the expression of Bax and p53. In addition, synephrine induces a significant decrease in the mRNA and protein levels of PI3K, AKT, and mTOR in H460 cells, followed by apoptosis induction [10].

Meanwhile, synephrine has no significant cytotoxic effect on immortalized normal esophageal epithelial cells and does not alter the levels of serum alanine transaminase (ALT) and aspartate transaminase (AST) in nude mice [90]. These data indicate that synephrine may provide a safer therapeutic profile.

However, a recent study shows that at a concentration of 25–5000 µM p-synephrine revealed no cytotoxic effect in human colon adenocarcinoma (Caco-2) cells. Furthermore, at the reduced concentration of 2–200 μM, synephrine stimulated ROS generation followed by an enhanced activity of the antioxidant enzymes GSH and CAT. In addition, synephrine activates MAPK1, the guanine nucleotide binding protein, the alpha-stimulating activity polypeptide (GNAS), the protein kinase cAMP-activated catalytic subunit alpha (PRKACA), and the AKT signaling pathway in Caco-2 cells [8]. More specifically, synephrine increases the expression of the PRKACA and the protein kinase cAMP-dependent type II regulatory subunit alpha (PRKAR2A) genes. These genes encode the catalytic subunits of PKA, the regulator of MAPK-dependent cell proliferation and differentiation [98,99]. In addition, synephrine significantly increases the expression of AKT1 (serine/threonine kinase 1) gene encoding the pro-proliferative signaling regulator. Furthermore, synephrine triggers GNAS upregulation, with subsequent production of the alpha subunits of the G protein, a key component in adenylate cyclase signaling and GPCR-dependent cellular responses [8]. These signaling pathways engage multiple targets for compounds including ephedrine, norepinephrine, and synephrine [100].

Studies in human hepatocellular carcinoma (HepG2) cell line demonstrate that the combination of synephrine and caffeine induces DNA damage and apoptosis at the concentrations of 3:60, 3:90, and 3:600 µM, respectively. These results are supported by the upregulation of genes associated with DNA repair and apoptosis, specifically caspase-9, and downregulation of the genes associated with cell cycle control [91]. However, treating HepG2 cells with synephrine causes the overproduction of intracellular ROS [92]. This could be related to a rapid transformation of synephrine by mitochondrial monoamine oxidases (MAOs) predominantly in liver cells as well as MAO-dependent ROS production [101,102]. The second possible mechanism is mediated by the stimulation of AMPKs [103], which act as the central regulators of energy metabolism [104]. Their excessive activation leads to an impaired electron transport in the respiratory chain and a subsequent mitochondrial overproduction of ROS [105]. Nevertheless, synephrine-induced ROS overproduction produces a cytostatic effect but does not cause significant DNA damage and does not induce chromosomal instability in HepG2 cells in vitro. These results indicate that synephrine causes acute oxidative stress; however, it is insufficient to sustain ROS overproduction over a period of time [91].

As mentioned above, synephrine is a stable hydrolysis product of CpdA [26], a SEGRA with well-known anti-cancer properties. The anti-cancer effects of both CpdA and GCs are partially mediated by apoptosis induction in cancer cells via the following processes: up-regulation of the pro-apoptotic members of the B-cell lymphoma (Bcl-2) family, such as Bim, Bid and Bad; suppression of the anti-apoptotic members, such as Bcl-2, Mcl-1, and Bcl-xL [93,106,107]; as well as inhibition of pro-proliferative signaling including AP-1 and NF-κB [16]. GCs (rather than CpdA) can also inhibit proliferation by suppressing c-MYC [108,109]. Furthermore, GR regulates the expression of multiple miRNAs, and, in particular, suppression of the miR-17-92 cluster correlates with apoptosis [110]. In contrast to CpdA, GCs have inhibited cell migration/invasion in in vitro models through a roster of mechanisms including downregulation of the Ras homolog family member A (RhoA) [111], matrix metalloproteinases 2 and 9 (MMP 2,9), and IL-6 [112], or inducing E-cadherin [113], which inhibits angiogenesis by suppressing the pro-angiogenic factors, including IL-8 and vascular endothelial growth factor (VEGF) [114] (Table 1).

### 5.3. The Effects on Diabetes Mellitus and Obesity

The anti-diabetic effect of p-synephrine has been described in the literature with nuance. Therefore, p-synephrine reduces glucose production by rat hepatocytes H4IIE in vitro, as well as the expression of glucose metabolism genes, glucose-6-phosphatase (G6Pase) and phosphoenolpyruvate carboxykinase (PEPCK) [115], at concentrations of 1-100 μM in a dose-dependent manner. Importantly, p-synephrine inhibits the activity of α-amylase, α-glycosidase, acetylcholinesterase, butyrylcholinesterase, and carbonic anhydrase [44]. Compounds with the described properties are considered for potential use in reducing postprandial blood glucose levels [115].

In the murine model of alloxan-induced diabetes mellitus, p-synephrine prevents alloxan-induced changes in body weight, organ parameters, serum uric acid, and serum creatinine levels as well as improves the lipid profile. The described effects are associated with the activation of enzymes of the antioxidant system, such as SOD and CAT, the inhibition of NF-κB and MAPK signaling, as well as changes in GSH content [7] (Table 1).

Importantly, the effect of p-synephrine on the pharmacodynamics and pharmacokinetics of gliclazide, a hypoglycemic agent from the group of sulfonylurea derivatives, has been described. The repeated administration of p-synephrine in the presence of gliclazide results in a significant decrease in blood glucose levels as well as an increase in insulin levels and an activation of pancreatic β-cell function in rats and rabbits [96]. The observed changes can be explained by the agonistic activity of synephrine on β3-AR, leading to an increase in the metabolic rate, as well as by synephrine’s ability to exhibit hypoglycemic and insulin-stimulating properties. p-Synephrine was shown to stimulate the translocation of the glucose-4 transporter protein from the cytoplasm to the cytoplasmic membrane, leading to a decrease in insulin resistance [116].

Furthermore, synephrine modulates obesity development and lipid metabolism disorders, and is widely used as a component of weight loss supplements [117]. Numerous studies have revealed the favorable therapeutic profile of p-synephrine: (1) an increase in resting metabolic rate and energy expenditure; (2) inhibition of glucose production [94]; (3) thermogenic activity [118]; (4) influence on enzymes activity [2,115]; (5) lipolytic activity; (6) catabolic activity; and (7) an influence on the differentiation of beige adipocytes [44].

Beige adipocytes are known to play an important role in increasing energy expenditure through differentiation. p-Synephrine was found to be an active compound that increases the level of the uncoupling protein 1 (UCP1) mRNA in stromal vascular fraction (SVF) cells cultured in beige adipocyte differentiation medium. p-Synephrine induces morphological changes specific to beige adipocytes in a dose-dependent manner. Similar effects were also observed in stromal vascular fraction cells obtained from leptin receptor-deficient db/db mice. These effects were mainly mediated by the activation of β3-AdR and, partially, insulin signaling, leading to the stimulation of beige adipocyte differentiation, UCP1 expression, and an increase in mTORC1 activity. In line with these findings, the combination of p-synephrine and espidulin, a flavone of natural origin, significantly inhibited expression of the genes encoding adipogenic proteins including MAPKs (ERK, ERK, JNK, and P38), C/EBPα, C/EBPβ, PPARγ, and GR [119]. In addition, p-synephrine regulates food intake and energy homeostasis, exerting effects on the central nervous system. Specifically, it activates neuromedin U2 receptor (NMU2R) in a dose-dependent manner [120]. NMUR2 is expressed in the hypothalamic regions of the brain and is involved in regulating energy balance, food intake, nociception, and stress [119]. All the data presented above demonstrate the potential of p-synephrine for preventing and treating obesity and related diseases [100].

Considering synephrine as an alternative to GR ligands, it is important to highlight the adverse effects of GCs on metabolic processes. The metabolic effects of GCs are associated with physiological mechanisms of peripheral insulin resistance, hyperglycemia, and dyslipidemia. In contrast, CpdA, a non-steroidal SEGRA, does not induce the expression of genes associated with metabolism and metabolic disorders [61,62,121,122]. Specifically, CpdA does not elevate hepatic gluconeogenesis, adipocyte lipolysis, or proteolysis, and causes an increase in the circulation of free fatty acids and lipid accumulation in skeletal muscles and liver, which leads to the development of insulin resistance in these tissues [123,124]. Direct GC-related insulin release inhibition is demonstrated in an in vivo study on transgenic mice overexpressing GR in β-cells [125]. The mechanisms of GC-dependent inhibition of insulin release in β-cells are likely to involve changes in the expression of TA-related subsets of genes important for glucose sensitivity and insulin secretion [121,126]. In contrast to GC, CpdA does not induce the stress-induced mTOR inhibitor DDIT4, a major molecular target of GC adverse effects including atrophic and metabolic changes [122,127,128,129,130]. Moreover, it has also been demonstrated that the targeted genetic deletion of DDIT4 through treatment with SEGRA such as ZK245186/mapracorate or with the inhibitors of DDIT4 expression shifts the function of GC-activated GR towards TR [122,128,129,130].

### 5.4. The Effect on the Cardiovascular System

Sympathomimetics vary widely in their ability to activate or inhibit different ARs. As was previously described, p-synephrine is chemically similar to sympathomimetic agents such as ephedrine. However, p-synephrine does not cause an increase in blood pressure or tachycardia, since it has a lower affinity for α1 and α2-AR, as well as β1 and β2-AR, compared with classic sympathomimetics [118,131].

In a recent placebo-controlled, double-blind study, it was shown that p-synephrin dramatically reduced diastolic blood pressure and mean arterial pressure when a person was in a quiet sitting position, while p-synephrine did not affect systolic blood pressure or heart rate. Adding p-synephrine to caffeine also did not increase systolic blood pressure or heart rate. This indicates that p-synephrine does not affect the cardiovascular system [132].

Another double-blind, placebo-controlled crossover study in 18 healthy subjects showed that p-synephrine does not lead to any significant changes in electrocardiograms, heart rate, systolic blood pressure, blood chemistry, or blood cell count in 30 min–8 h and thereafter within 15 days after the treatment [28]. Furthermore, the results of both published and unpublished clinical trials, involving approximately 360 subjects, show that p-synephrine alone or in combination with caffeine neither appears to cause significant side effects on the cardiovascular system nor poses a risk to human health when it is used in doses normally taken orally [54,133].

Importantly, GCs-induced side effects on the cardiovascular system are mediated by interactions with ARs. GCs increase β-adrenergic reactivity [83]. In lung cells, corticosteroids could increase the number of β2-receptors [134]. ARs such as α-AR and β-AR are involved in the regulation of cardiovascular functions. α-AR activation leads to an elevation in blood pressure and heart rate, whereas β-AR activation increases cardiovascular function and oxygen demand. CpdA’s effects on the cardiovascular system have not yet been described in the pertinent literature.

## 6. Conclusions

In this review, we have described a wide range of biological effects of synephrine, its identified and potential molecular targets, and its possible mechanisms of action. In addition, we have compared the molecular targets and effects of synephrine, a selective glucocorticoid receptor agonist (SEGRA) CpdA, with molecules with a similar chemical structure, as well as classic GCs. We have also considered synephrine as potentially useful in revealing SEGRA properties, when using it as the template for synthesis of putative SEGRAs. The studies on SEGRAs demonstrate that SEGRAs, including CpdA, could be effective alternatives to GCs in the treatment of inflammatory diseases, cancer, metabolic, and cardiovascular diseases, using CpdA as an example. However, the degradation of CpdA can lead to toxic metabolites being formed. Therefore, searching for synephrine derivatives, our overview here can open up new avenues for solving this problem. In addition, this review provides information on the safe pharmacological profile of synephrine. Overall, due to these properties, synephrine and its potential derivatives are considered to be promising for further preclinical studies.

## Figures and Tables

**Figure 1 ijms-24-17537-f001:**
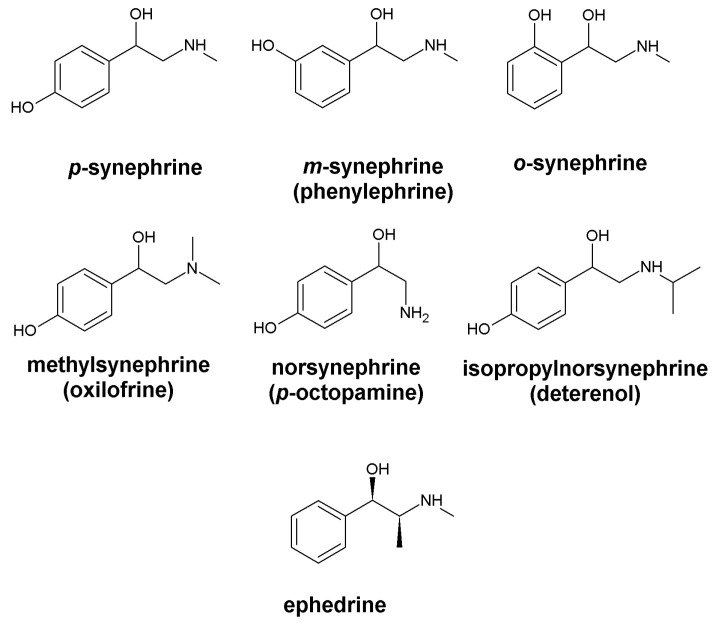
Structures of the synephrine isomers, derivatives, and ephedrine.

**Figure 2 ijms-24-17537-f002:**
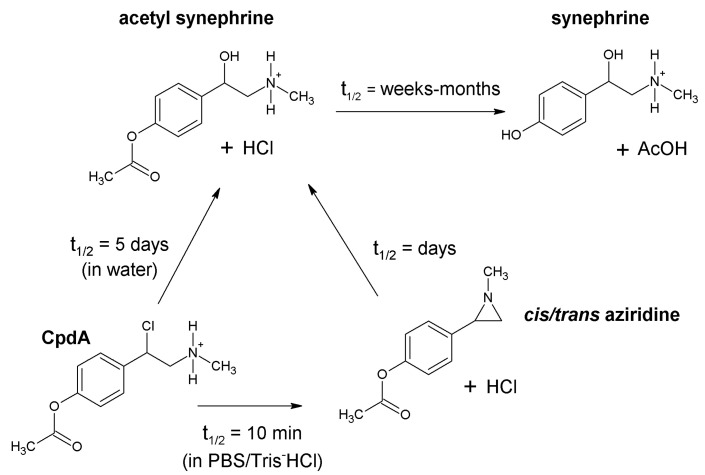
Decomposition of CpdA to aziridine derivatives and synephrine depends on the solvent and pH [22].

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
