# Peer review of "Synephrine and Its Derivative Compound A: Common and Specific Biological Effects"

_ijms, 2023, doi:10.3390/ijms242417537_

Round 1

Reviewer 1 Report

Comments and Suggestions for Authors

My opinion on the draft article titled "Synephrine and its derivative CpdA: common and specific biological effects" is that it is a well-designed study in terms of being a very current topic and providing detailed information to the authors.

The work needs minor revisions to be made. It is suitable for publication afterwards.

First of all, I saw that the article draft did not include a recent study that was relevant to the subject. Since this is a review, I believe it should be included in the relevant publication (Perova, I. B., Eller, K. I., Musatov, A. V., & Tymolskaya, E. V. (2021). Synephrine in dietary supplements and specialized foodstuffs: biological activity, safety and methods of analysis. Voprosy Pitaniia , 90(6), 101-113).

There are places in the article draft where italics are omitted, especially when writing species names, and these parts should be corrected.

There are typos and font differences in the references section. It should be edited following the journal rules.

I think Table 1 needs to be rearranged. It should be made simpler and more understandable.

Author Response

We thank the reviewer for the careful and thorough comments. Below we provide point-by-point responses.

First of all, I saw that the article draft did not include a recent study that was relevant to the subject. Since this is a review, I believe it should be included in the relevant publication (Perova, I. B., Eller, K. I., Musatov, A. V., & Tymolskaya, E. V. (2021). Synephrine in dietary supplements and specialized foodstuffs: biological activity, safety and methods of analysis. Voprosy Pitaniia , 90(6), 101-113).

We added the necessary information (new reference 132, highlighted in yellow).

There are places in the article draft where italics are omitted, especially when writing species names, and these parts should be corrected.

We made all the necessary corrections.

There are typos and font differences in the references section. It should be edited following the journal rules.

We made all the necessary corrections

I think Table 1 needs to be rearranged. It should be made simpler and more understandable.

The table is made in accordance with the journal's requirements for the design of tables.

Reviewer 2 Report

Comments and Suggestions for Authors

Review

Synephrine and its derivative CpdA: common and specific biological effects

A brief summary

The topic is interesting and in keeping with current trends. The Review is properly designed and conducted, the effect interesting. The authors seem to have learned the issue and know how to share this knowledge with their readers. However, amendments and corrections are required.

Broad comments

1. The structures of Oxilofrine and Deterenol need to be corrected.

2. A compilation in a table of all abbreviations and acronyms used would be nice.

3. Latin terms, such as “in vitro”, “Rutaceae” or “Citrus aurantium”, should be written in italics.

4. Typos need to be corrected, spaces inserted or removed, etc.

Author Response

We thank the reviewer for the careful and thorough comments. Below we provide point-by-point response (in the manuscript all corrections are highlighted in yellow).

1. The structures of Oxilofrine and Deterenol need to be corrected.

The structure of the substances has been adjusted

2. A compilation in a table of all abbreviations and acronyms used would be nice.

A list of abbreviations has been added at the end of this manuscript.

3. Latin terms, such as “in vitro”, “Rutaceae” or “Citrus aurantium”, should be written in italics.

We made the necessary corrections (highlighted in yellow)

4. Typos need to be corrected, spaces inserted or removed, etc.

We made the necessary corrections (highlighted in yellow)

Reviewer 3 Report

Comments and Suggestions for Authors

General comment

The manuscript ''Synephrine and its derivative CpdA: common and specific bio-logical effects" described synephrine (origin, chemical composition, receptors, pharmacological properties), phytochemical compund found in bitter orange and some other medicinal plants which is used as a dietary supplement for weight loss reduction. Also, the comparison of synephrine and its selective glucocorticoid receptor agonist CpdA was conducted.

 Article is well done and I have only minor comments.

Abstract

Last line 34: omit space before ''cological''

Keywords

Think to show ''SEGRA'' as a separate keyword

1. Introduction

Paragraph 2, line 2: put ''Citrus aurantium'' into Italic

Paragraph 2, lines 2, 3: what is ''Zhi Shi'' and ''Kijitsu''? Cultivars of Citrus aurantium? If yes, write in the scientific correct way. The same goes for ''Nova tangerines'' and ''Marrs''.

2. Source, chemical and pharmacological features of synephrine

Figure 1 and page 3, paragraph 4: ''p-octopamine'' or ''p-octamine''?

Page 3, paragraph 3: ''m-Synephrine'' or ''m-synephrine''? I suggest to use the same style of writting.

Page 3, paragraph 5: put ''Citrus aurantium'' into Italic

3. Mechanism of synephrine effects

Page 4, paragraph 1: ''β-Agonists'' or ''β-agonist''? I suggest to use the same style of writting.

Page 4, paragraph 2: ''p-Synephrine'' and ''m-synephrine''. I suggest to use the same style of writting.

4. Chemical and pharmacological features of CpdA and its target receptors

Page 5, paragraph 2: put ''Salsola tuberculatiformis'' into Italic

Page 6, paragraph 2: font color?

Page 11, paragraph 1: omit space between ''to'' and ''sustain''

5.3. The effects on diabetes mellitus and obesity

Page 11, paragraph 2: omit space before ''as wel''

Page 11, paragraph 3: ''p-Synephrine'' or ''p-synephrine''

Page 12, paragraph 2: ''p-Synephrine''?

Page 12, paragraph 3: add dot before ''Direct''

References

Check the references writing style again. For example, use the same style for writing title of references (compare ref. 93 and 94)

Ref. no. 26, 29, 35: change ''Citrus Aurantium'' into ''Citrus aurantium''

Ref. no. 28: change ''p -Synephrine'' into ''p-Synephrine''

Ref. no. 31: change ''p -octopamine'' into ''p-octopamine'' and ''m -synephrine'' into ''m -synephrine''

Ref. no. 41, 66: change uppercase into lowercase

Author Response

We thank the reviewer for the careful and thorough comments. Below we provide point-by-point response (in the manuscript all corrections are highlighted in yellow).

1. Abstract
Last line 34: omit space before ''cological''

It refers to the word ''pharmacological'' (highlighted in yellow).

2. Keywords

Think to show ''SEGRA'' as a separate keyword

SEGRA is the abbreviation for selective glucocorticoid receptor activator, specific keyword from the list

3. Introduction

Paragraph 2, line 2: put ''Citrus aurantium'' into Italic

We made the necessary corrections (highlighted in yellow).

Paragraph 2, lines 2, 3: what is ''Zhi Shi'' and ''Kijitsu''? Cultivars of Citrus aurantium? If yes, write in the scientific correct way. The same goes for ''Nova tangerines'' and ''Marrs''.

The names of the plants are specified in the text of the manuscript.

4. Source, chemical and pharmacological features of synephrine

Figure 1 and page 3, paragraph 4: ''p-octopamine'' or ''p-octamine''?

We corrected it to ''p-octopamine'' (highlighted in yellow).

Page 3, paragraph 3: ''m-Synephrine'' or ''m-synephrine''? I suggest to use the same style of writting.

In this case, throughout the entire text of the manuscript, a unified approach to writing names is used: if the name is used at the beginning of a sentence, then the spelling is used ''m-Synephrine'', if inside the sentence, then the spelling is used ''m-synephrine''.

Page 3, paragraph 5: put ''Citrus aurantium'' into Italic

Done.

5. Mechanism of synephrine effects

Page 4, paragraph 1: ''β-Agonists'' or ''β-agonist''? I suggest to use the same style of writting.

Same comments as for synephrine above.

Page 4, paragraph 2: ''p-Synephrine'' and ''m-synephrine''. I suggest to use the same style of writting.

Same comments as for synephrine above.

6. Chemical and pharmacological features of CpdA and its target receptors

Page 5, paragraph 2: put ''Salsola tuberculatiformis'' into Italic.

Done.

Page 6, paragraph 2: font color?

Same font color throughout the manuscript (black).

Page 11, paragraph 1: omit space between ''to'' and ''sustain''

Done.

The effects on diabetes mellitus and obesity
Page 11, paragraph 2: omit space before ''as wel''

Done.

Page 11, paragraph 3: ''p-Synephrine'' or ''p-synephrine''

Same comments as for synephrine above.

Page 12, paragraph 2: ''p-Synephrine''?

Same comments as for synephrine above.

Page 12, paragraph 3: add dot before ''Direct''

Done.

7. References
Check the references writing style again. For example, use the same style for writing title of references (compare ref. 93 and 94)
Ref. no. 26, 29, 35: change ''Citrus Aurantium'' into ''Citrus aurantium''
Ref. no. 28: change ''p -Synephrine'' into ''p-Synephrine''
Ref. no. 31: change ''p -octopamine'' into ''p-octopamine'' and ''m -synephrine'' into ''m -synephrine''
Ref. no. 41, 66: change uppercase into lowercase

Done.